# Relationships between Plantar Pressure Distribution and Rearfoot Alignment in the Taiwanese College Athletes with Plantar Fasciopathy during Static Standing and Walking

**DOI:** 10.3390/ijerph182412942

**Published:** 2021-12-08

**Authors:** Tong-Hsien Chow, Yih-Shyuan Chen, Chin-Chia Hsu

**Affiliations:** 1Department of Leisure Sport and Health Management, St. John’s University, New Taipei 25135, Taiwan; thchow1122@mail.sju.edu.tw (T.-H.C.); hill@mail.sju.edu.tw (C.-C.H.); 2Department of Education, National Pingtung University, Pingtung 900391, Taiwan

**Keywords:** plantar fasciopathy (PF), plantar pressure distributions (PPDs), arch index (AI), rearfoot postural alignment, pronated low-arched foot

## Abstract

Background: Plantar fasciopathy (PF) is usually related to changes in foot arch, foot shape and rearfoot posture. However, little research has been implemented by using large-scale datasets, and even less has been conducted centering on plantar pressure distributions (PPDs) of different genders of PF athletes. This study aimed to investigate the relationships among the arch index (AI), the PPDs and the rearfoot postural alignment in hundreds of college athletes with PF during static standing and walking. Methods: Cross-sectional study of 100 male and 102 female athletes with PF was undertaken. The PF athletes’ pain assessment and self-reported health status were examined for evaluating their musculoskeletal painful areas. Results: The PF athletes’ PPDs mainly concentrated on inner feet in static standing, and transferred to lateral forefeet during the midstance phase of walking. The males’ PPDs from the static standing to the midstance phase of walking mainly transferred to anterolateral feet. The females’ PPDs mainly transferred to posterolateral feet. The PF athletes’ static rearfoot alignment matched the valgus posture pattern. The medial band of plantar fascia and calcaneus were the common musculoskeletal pain areas. Conclusions: Characteristics of higher plantar loads beneath medial feet associated with rearfoot valgus in bipedal static stance could be the traceable features for PF-related foot diagrams. Higher plantar loads mainly exerted on the lateral forefoot during the midstance phase of walking, and specifically concentrated on outer feet during the transition from static to dynamic state. Pain profiles seem to echo PPDs, which could function as the traceable beginning for the possible link among pronated low-arched feet, PF, metatarsalgia, calcanitis and Achilles tendinitis.

## 1. Introduction

Plantar fasciopathy (PF) is considered to be the third most prevalent musculoskeletal disorder in runners [1,2,3] and characterized by pain at the plantar fascia insertion point [4]. Previous study explored the pathogenesis of PF from the athlete’s specific internal and external risk factors [5]. Several congenital factors for the development of PF in runners have been confirmed to be related to changes in foot shape [6], rearfoot valgus posture [7,8], and abnormal foot arch structure [1,8,9]. Many studies have shown that changes in medial longitudinal arch (MLA) geometry (lower [7] or higher [9,10]) and the presence of pain in runners with PF may cause increased plantar load [7,8]. It has been widely accepted that the combination of MLA height and rearfoot valgus angles serves as an effective predictor of pressure-time integral in the midfoot and rearfoot regions of healthy runners [11].

Most healthy runners with a higher MLA are characterized with a higher vertical load rate [12,13] and peak pressure on the rearfoot during running [13]. Yet, people with a lower MLA associated with increased rearfoot pronation generally have a greater pressure load on the medial side of the calcaneus [14,15,16], and this usually results in greater stretching of the plantar fascia [3,7,17]. A valgus alignment of the rearfoot or pronated foot posture is highly likely to increase the possibility of the generalized foot pain [18]. Di Caprio et al. pointed out the high proportion of the high arch runners suffering from PF. This is because the elevated MLA of runners’ feet may cause excessive stretching of the plantar fascia, and this, in turns, contributes to reduced tissue flexibility [9]. In the long term, the ability of the foot to cushion the ground reaction force may decrease, which may impose more mechanical stress on the calcaneus [17], thereby affecting the dynamic foot function of the human body [18]. However, some cases occur in people with elevated MLA and valgus rearfoot, which easily leads to microtrauma and microtearing of the plantar fascia, resulting in the inflammation characteristic of the acute phase [19,20]. Based on the related studies, individuals with PF promoted changes in foot roll-over patterns due to painful stimuli, and this tended to cause a decrease in the load on the rearfoot and an increase in the load on the midfoot [21], forefoot [22] and toes [22,23] due to the protective mechanisms of pain.

Given the above context, there seems to be a potential risk of PF between the MLA geometry and the rearfoot valgus posture, and most studies have focused on runners. Therefore, it can be argued that it is crucial to conduct the in-depth study of exploring the changes in static and dynamic plantar pressure distribution concerning the rearfoot posture in PF individuals, in terms of providing valuable information on healthy athletes’ foot care and prevention prescriptions, injured athletes’ foot correction and rehabilitation, and orthotics design, such as insoles and motion-control shoes for PF patients [24].

According to the studies, people with PF may trigger the foot over rolling due to the pain and cause changes in plantar pressure distribution [21,22,23]. The causal relationship between the etiology of the plantar fasciopathy caused by the specific plantar pressure distribution and the special plantar pressure pattern induced by the plantar fasciopathy pain is worth attention and exploration. On this basis, one of the key aims of this study was to establish the overall reliability of the relationships among the arch index (AI), the plantar pressure distributions (PPDs) and the rearfoot postural alignment by using the large-scale datasets of the college athletes with PF during static standing and walking. The other research aim of this study was to ascertain the relationships of the lower extremity musculoskeletal pain caused by PF by means of examining the correlation between the lower limb pain profiles and PPDs.

## 2. Methods

### 2.1. Participants

This cross-sectional study examined the relationships among the PF, the PPDs, the rearfoot postural alignment and the potential pain profiles of the lower extremity. Research participants within this study comprised 411 college and university students of both sexes in Taiwan, and were categorized into two groups: 202 college athletes with diagnoses of PF (the PF group) and 209 healthy age-matched college students (the control group). In the present study, the inclusion criteria in the PF group (100 males and 102 females) were the athletes with diagnoses of bilateral symptomatic medio-plantar heel pain for more than four months based on the physician’s certificate, clinical examination and ultrasound images [24]. They were recruited from St. John’s University, National Taiwan Sport University, National Taiwan University of Sport, College of Kinesiology in the University of Taipei, National Taiwan Normal University, Chinese Culture University, Ching Kuo Institute of Management and Health, and seven city sports centers in Taipei, Taiwan. The clinical examination of PF symptoms was conducted by checking the pain which occurred during palpation of the plantar fascia after waking up in the morning, while remaining in the standing position, when taking the first few steps, while sitting for long periods of time, and after physical activity [22,23]. In this study, athletes in the PF group were the qualified first or second-division players from different areas of sports expertise. Definition of the qualified first or second-division players was the athletes who had more than four successive years of sporting experiences in competition and were diagnosed with PF confirmed by a clinical examination. The PF group’s exercise workout schedules were at least 4 days weekly, including 2-h physical and shooting training, 2-h basic movements and tactical training, 2-h weight training and 1–2 h of high-intensity interval individual special sports training. 

The control group (105 males and 104 females) was constituted of healthy college students. The exclusion criteria in both groups were a history of previous surgery in the lower limbs, traumas or fractures of the lower limbs in the previous six months, leg length discrepancies and other musculoskeletal disorders, such as neuropathies, rheumatoid arthritis and calcaneal spurs. In addition, many studies have noted the influence of body weight on the characteristics of foot arch and plantar pressures as can be evidenced by the related research which verified the strong link between obesity and flat feet in adults and children [25,26,27,28]. Considering the effect of body weight on shape characteristics of the foot, each participant’s body mass index (BMI) within this study was required to be between 18.5 and 22.9. This particular range of BMI is defined by the World Health Organization (WHO) and Asia-Pacific guidelines as normal healthy weight [29,30]. All participants’ age, sex, height, body weight and BMI values were shown in Table 1.

Before the experiments, all participants were required to sign the informed consent forms for participation in this study. All experiments in this study followed the guidelines of the research ethics committee of National Taiwan University and the recommendations of the Declaration of Helsinki.

### 2.2. Instruments and Equipment

The footprint was acquired via the JC Mat optical plantar pressure analysis system (View Grand International Co Ltd, New Taipei City, Taiwan), which was used for measuring the static and dynamic PPDs. The repeatability and reproducibility of the device were confirmed in previous studies [31,32]. The measurement technology and principles of the JC Mat were parallel to the operation principles of the Harris footprint measurement. The standing platform of the JC Mat contained 25 sensors in each square centimeter; that is, the gross sensors on each side (32 × 17 cm) of the standing platform were 13,600. The sensitive pressure sensing with a wide working area displayed and marked the delicate plantar pressure image with round dots. Thus, the pressure profiles from footprints and barefoot images were captured instantly, and the built-in FPDS-Pro software (View Grand International Co Ltd, New Taipei City, Taiwan) was qualified for analyzing the arch index (AI), the PPDs, the balance of the center of gravity, the toe angles, the footprints and 3D foot images of both feet.

### 2.3. PPDs Assessment

It took one year to recruit and measure the research participants. In order to ensure the consistency and credibility of this study, experimental time for each participant was controlled between 2 and 5 pm on the same day of the week. All participants were instructed to measure their body weight and height for calculating the BMI values and record their self-reported health data for identifying possible painful areas. In order to ascertain the static footprint data via the brief trials of the static upright standing, the participants were asked to obey the following steps:Roll both trouser legs up to above the knees to prevent the clothing from limiting movements of the extremities;Stand barefoot on the sensing cushion with the specific marks and measuring range of the JC Mat;Relax the body. Control and balance the center of gravity by standing with feet shoulder-width apart and with body weight evenly distributed on feet;Stampede for six to eight steps, then stand still with a natural posture and arms hanging straight down at the sides;Face the experiment instructor. Look the instructor straight in the eye. Keep the body stationary and balanced until no obvious changes are seen in the foot pressure value measured by the JC Mat.

When the participants reached the condition in step 5, pressure profiles from the static footprints were recorded directly. In the follow-up measurement of the dynamic footprints, the participants walked with a natural gait speed on a 4-meter-long walkway with JC Mat embedded. Multiple walking trials were completed until at least three steps for each foot were correctly acquired (i.e., the sensing cushion with marks of the specific measuring range of the JC Mat was struck with a single foot).

### 2.4. PPDs Data Analysis

The PPDs of static standing and walking footprints were explored in three regions and six subregions of both feet by the computer program (FPDS-Pro software). After capturing the footprint image, the software generated the first line (a perpendicular line) on the footprint image. The perpendicular line extended from the tip of the second toe to the center of the heel, and was drawn tangential to the most anterior and posterior part of the footprint excluding the toes. The software formed four parallel lines perpendicular to the first line and divided the footprint into three equal regions (regions A, B, and C) and six subregions (subregions 1–6). Regions A, B, and C were defined as the forefoot, midfoot, and rearfoot regions, respectively. Subregions 1 to 6 were defined as the lateral metatarsal bone (LM), lateral longitudinal arch (LLA), lateral heel (LH), medial metatarsal bone (MM), medial longitudinal arch (MLA), and medial heel (LH), respectively. The AI ratio method proposed by Cavanagh and Rodgers [33] assumes that the AI is calculated as the ratio of the area of the middle third of the footprint divided by the entire footprint area excluding the toes, i.e., AI = B/(A + B + C). For Cavanagh and Rodgers [33], an AI lower than 0.21 is classified as a high-arched foot, 0.21 to 0.26 is classified as a normal arched foot, and higher than 0.26 is classified as a flat-arched foot.

### 2.5. Rearfoot Postural Alignment Assessment

After the PPDs assessment, each participant’s posterior view of the rearfoot postural alignment was examined. All participants were guided to stand over a 30 cm height platform and keep their feet a natural width apart (about 12–15 cm). The posterior view of each participant’s rearfoot postural alignment image was obtained (with a minimum size of 754 pixels and 96-ppi screen resolution) with a digital camera (Figure 1). According to the literature by Ribeiro et al. [24], the method of calculating the rearfoot static angle was as follows: to confirm that the rearfoot of both feet relaxed stand on the same horizontal line, and to determine the anatomical points in the lower back area of the legs: (1) the posterior calcaneal tuberosity; (2) the second point above the center of the calcaneus; and (3) the lower third of the leg. In the 3-point connection, two lines will be automatically generated by the Biomech 2019-postural analysis software (Loran Engineering SrL, Castel Maggiore BO, Italy). The first standard straight of the lower extremity (a solid line) was drawn, which originated from the lower third of the leg to the calcaneal center. The second flip angle line of the lower extremity (a dotted line) was drawn, which originated from the posterior calcaneal tuberosity to the center of the calcaneus (Figure 2). The static rearfoot alignment was measured from the frontal alignment of a digital image by the software. The intersection of the extensions of both straight lines resulted in angles, which were classified as a normal foot (0° to 5°), varus (<0°), or valgus (>5°) [34].

### 2.6. Self-Reported Health Status Examination

Prior to the assessment of soft-tissue pain and skeleton arrangement, the self-reported health status of the pain level of the subjects’ feet with PF symptoms on return to weight-bearing following rest was assessed by a 10 cm Visual Analogue Scale (VAS) [35]. VAS is a valid and reliable tool and has been widely used in the PF-related studies [22,23]. All PF athletes rated their pain levels on VAS ranging from none to unbearable. Questionnaires were used for collecting research participants’ personal data. This process was mainly used for securing the qualified subjects’ data suitable for the present research.

### 2.7. Pain Assessment

The PF athletes’ soft-tissue pain and skeleton arrangement assessments were undertaken by a physiotherapist at the rehabilitation department of Tri-Service General Hospital in Taipei. These steps were essential for ensuring that the target subjects met the criteria and confirming their physiological symptom and pain locations. In the process of pain assessments, the physiotherapist examined the PF athletes’ self-reported health status questionnaires, medical history, physical activities and foot pain symptoms. Palpating examination of the PF symptoms was conducted as follows:Plantar pain reproduced by palpating the plantar medial calcaneal tubercle at the site of the plantar fascial insertion on the heel bone;Plantar pain reproduced with passive dorsiflexion of the foot and toes;Passive dorsiflexion of the first metatarsophalangeal joint (test to provoke symptoms at the plantar fascia by creating maximal stretch), positive test if the pain is reproduced.

In the process of the soft-tissue pain and skeleton arrangement assessments, the lower-limb pain was defined as the musculoskeletal pain which occurred during the past month and originated from the structures of the foot, ankle, knee, lower leg, and thigh. The definition excluded intermittent cramps, dermatologic conditions, digital calluses, and nighttime paresthesia from the analysis. A standardized protocol of the questioning and examination techniques was used for ensuring the precise nature of the complaint. Physical examinations were conducted to evaluate the PF athletes’ frequent soft-tissue and bone pains in their lower limbs based on the following steps:The physiotherapist examined the PF athletes’ self-reported health status and pain complaints, and guided them to stand with bare feet and roll their trouser legs up to above the knees.The physiotherapist examined the PF participants’ lower extremities by palpating and pressing their feet (including navicular bones, cuboid bones, phalanges, metatarsals, and calcaneus), ankles, patella, knees, hips, tibias, fibulas and femur according to the participants’ self-reported health status, and re-examined the corresponding position on the other side of the pain areas. The physiotherapist, then, assessed the skeletal arrangement of the participants’ lower limbs.In order to confirm the participants’ pain areas precisely, the physiotherapist examined the following specific parts of the participants’ common pain areas: (1) soft tissues, e.g., the plantar fascia, the Achilles tendon, the gastrocnemius, the tibialis anterior and posterior, the biceps, the quadriceps femoris, the medial and lateral ankle ligaments, the anterior cruciate ligaments, the medial and lateral collateral ligaments, the abductor hallucis and abductor digiti minimi of plantar plate and the lower back; (2) bone tissues of both feet, i.e., navicular bones, cuboid bones, phalanges, metatarsals, and calcaneus; (3) the ankles; (4) the patella; (5) the knees; (6) the hips and (7) tibias.

### 2.8. Statistical Analysis

Descriptive statistics were used for outlining research participants’ age, height, weight, and BMI values. Numerical data in this study were presented as mean ± SD. Parameters gained from the plantar pressure measurement regarding the AI values and six distinct subregional PPDs were used for comparison between two groups via the independent sample *t*-test. The six distinct subregional PPDs of PF athletes were compared between the static standing and the midstance phase of walking via the paired-samples *t*-test. Statistical significance was defined as *p* < 0.05 and *p* < 0.01. The statistical software program (IBM SPSS Statistics 20.0 statistical computing software,) (IBM SPSS Statistics 20; Chicago, IL, USA) was used for managing the statistical analyses.

## 3. Results

### 3.1. Arch Index

Compared with the control group, the results showed that the average bipedal AI value of the male PF group had a significantly higher trend. The results indicated that the male PF group’s arch type fell into the category of the low arch (Table 2).

### 3.2. PPDs Assessment in Static Standing

The PPDs were presented as percentages of the relative loads. The results showed that the PF athletes’ relative loads in static standing were mainly concentrated on the inner part of the entire foot. Compared with the control group, the PF group’s relative loads of the six subregions were significantly higher at the medial metatarsals (left foot: 25.33 ± 4.97%; right foot: 26.08 ± 5.30%; *p* < 0.01) and the medial longitudinal arches (left foot: 1.12 ± 0.36%; right foot: 1.36 ± 0.30%; *p* < 0.01) of both feet. The PF group’s relative loads of the medial heel (20.20 ± 5.78%; *p* < 0.05) of the left foot were much lower at the lateral metatarsals (left foot: 21.27 ± 5.29%; right foot: 22.90 ± 8.06%; *p* < 0.05), the lateral longitudinal arches (left foot: 18.58 ± 4.19%; right foot: 17.50 ± 4.51%; *p* < 0.05) and the lateral heels (left foot: 13.51 ± 5.25%; right foot: 12.06 ± 7.92%; *p* < 0.01) of both feet (Figure 3).

The male PF group’s relative loads were found to be higher at the medial metatarsals (left foot: 26.21 ± 4.67%; right foot: 27.49 ± 5.32%; *p* < 0.05) and the medial heels (left foot: 17.88 ± 6.52%; right foot: 19.24 ± 5.63%; *p* < 0.01) of both feet, and the medial longitudinal arch (1.37 ± 0.31%; *p* < 0.01) of right foot. They, however, were lower at the medial longitudinal arch (0.87 ± 0.02%; *p* < 0.01) of the left foot, the lateral longitudinal arch (16.98 ± 4.67%; *p* < 0.01) and the lateral heel (18.91 ± 5.67%; *p* < 0.01) of the right foot (Figure 4).

The female PF group identically reflected upon the higher relative loads at the medial metatarsals (left foot: 24.46 ± 5.13%; right foot: 24.69 ± 4.91%; *p* < 0.05) and the medial longitudinal arches (left foot: 1.37 ± 0.37%; right foot: 1.35 ± 0.29%; *p* < 0.01) of both feet, and the medial heel (20.97 ± 3.54%; *p* < 0.01) of the right foot; yet, they had lower relative loads at the lateral longitudinal arches (left foot: 18.21 ± 3.74%; right foot: 18.00 ± 4.32%; *p* < 0.05) of both feet and the lateral heel (9.78 ± 2.44%; *p* < 0.01) of the right foot (Figure 5).

### 3.3. PPDs Assessment during the Midstance Phase of Walking

Findings from the midstance phase of walking indicated that the PF athletes’ relative loads were mainly transferred to the lateral part of the forefoot. Compared with the control group, the PF group’s relative loads of the six subregions were exerted more on the lateral metatarsals (left foot: 29.62 ± 4.01%; right foot: 27.76 ± 5.84%; *p* < 0.05) of both feet. Nonetheless, the PF group’s relative loads were found to decrease at the medial metatarsal (24.19 ± 5.04%; *p* < 0.01), the medial longitudinal arch (0.81 ± 0.51%; *p* < 0.05) and the medial heel (7.48 ± 3.82%; *p* < 0.05) of the left foot, and the lateral heels (left foot: 16.77 ± 2.01%; right foot: 17.95 ± 5.21%; *p* < 0.01) of both feet (Figure 6). 

In the male PF group, the relative loads were mainly focused on the lateral metatarsals (left foot: 30.42 ± 3.18%; right foot: 26.74 ± 4.96%; *p* < 0.01) of both feet. Yet, they were relatively low at the lateral heels (left foot: 16.39 ± 2.31%; right foot: 16.79 ± 4.48%; *p* < 0.05) and the medial heels (left foot: 6.63 ± 3.75%; right foot: 9.09 ± 6.14%; *p* < 0.01) of both feet, and the medial longitudinal arch (1.02 ± 0.49%; *p* < 0.05) of the right foot (Figure 7).

With regard to the female PF group, the relative loads were mainly exerted on the lateral metatarsals (left foot: 28.85 ± 4.57%; right foot: 28.75 ± 6.47%; *p* < 0.05) of both feet, whereas were found to be lower at the medial longitudinal arches (left foot: 0.77 ± 0.61%; right foot: 0.94 ± 0.65%; *p* < 0.05) and the lateral heels (left foot: 17.13 ± 1.59%; right foot: 19.08 ± 5.63%; *p* < 0.05) of both feet, and the medial heel (6.57 ± 2.70%; *p* < 0.05) of the right foot (Figure 8). 

### 3.4. PPDs of the Static Standing Versus the Midstance Phase of Walking 

The results of the PPDs changes from the static standing to the midstance phase of walking in the PF athletes showed that the relative loads were considerably focused on the lateral part of the foot. Findings from the male PF group showed that their relative loads were mainly transferred to the lateral metatarsals of both feet and the lateral longitudinal arch of the left foot. The female PF group revealed that their plantar loads were mainly transferred to the lateral metatarsal of the left foot, the lateral longitudinal arches and the lateral heels of both feet. It can be summarized from the findings by saying that the male PF athletes’ relative loads were mainly transferred to the anterolateral part of the foot. Yet, the female PF athletes’ relative loads were mainly transferred to the posterolateral part of the foot (Table 3).

### 3.5. Rearfoot Postural Alignment Assessment

With regards to the changes in the bipedal rearfoot angle, the findings showed that the values of the static rearfoot alignment in the PF athletes, particularly the male PF group, conformed to the valgus posture pattern in comparison with the control group (Table 4).

### 3.6. Self-Reported Health Status and Pain Assessment 

The PF athletes reported a median symptom duration of six months (range, 2–48 months), and rated their pains of the symptomatic feet on the VAS of 2 to 6 cm. The assessing results pointed out the fact that the top 10 common bone pain areas and soft tissue pains in the PF athletes in terms of the proportional order were listed in Table 5.

## 4. Discussion

Much research has focused on the plantar loads of PF athletes. Yet, at present, little has been implemented by using large-scale datasets, and even less research has been conducted centering on the differences in plantar pressure distributions between age-matched male and female PF athletes. In order to expand our knowledge of the relationships among the AI, the PPDs, the rearfoot postural alignment, and the pain profiles which caused by PF, the present study examined the causal relationships among these specific factors from hundreds of age-matched PF athletes of both genders during static standing and in midstance phase of walking as well. In this study, the recruited participants’ BMI values were within the normal range, considering weight gains are generally viewed as the key determinant of changes in the shape of the arch [36]. Notably, the AI value used for defining the foot type category within this study was based on the literature by Cavanagh and Rodgers which observed that the AI value of normal arches was in the 0.21–0.26 range [33]. Nevertheless, findings from the present study were slightly different from those presented by Cavanagh and Rodgers. The difference was most likely explained by the fact that the sample characteristics, research contexts and approaches within this study were different from those applied in Cavanagh and Rodgers’ research. In their research, Cavanagh and Rodgers highlighted the critical value of AI (i.e., a mean AI of 0.23) calculated in the context from 107 young adults (mean age 30 years) without any foot symptom. In the present study, the critical value of static AI of both feet in the control group was ranged 0.22–0.23 (male: 0.23 to 0.24; female: 0.21 to 0.22), which was calculated from average of 209 college and university students of both sexes in Taiwan (mean age 20 years) with normal BMI (22.3 ± 1.2) and without musculoskeletal disorders of extremities. The difference in the normal range may inevitably exist in the subjective judgments of researchers, assessing equipment, characteristics of subjects and sample size; nonetheless, the study was set within the identical research condition. Based on the findings, the AI value of both feet in the male PF group was between 0.26 and 0.27, which was much higher than that of male controls. Therefore, the male PF group in the present study could be classified as the low-arched foot. The results seem to echo the related study which revealed that runners with a history of PF were generally connected with the anatomical factors of the lower medial longitudinal arch [7]. The PF-related pain was considered to be correlated with the midfoot loading in the symptomatic foot [23]. Individuals with the PF symptomatic foot may experience an unstable foot rolling process due to pains and the increased plantar loads located in the midfoot [21], forefoot [21] and toes [21,23]. 

From the results of the entire plantar pressure distribution, the PF athletes’ relative loads in bipedal static stance were mainly concentrated on the inner side of the entire foot; hence, their relative loads on the outer side was relatively low. In the aspect of the rearfoot postural alignment measurement, the PF athletes’ bipedal rearfoot angle in static state were conformed to the rearfoot valgus posture. The pattern of the results seemed to constitute a pronation foot feature. Furthermore, the results corresponded to previous studies which maintained that an increase in the rearfoot pronation associated with a lower medial longitudinal arch could result in higher plantar loads beneath the medial part of rearfoot [15,16], and that this specific syndrome tended to lead to a greater stretch to the plantar fascia [3,17]. A valgus alignment of the rearfoot or pronated foot posture obviously increased the likelihood of the generalized foot pain [18]. People with PF usually compensate and control pains by exhibiting greater total rearfoot valgus, increasing the medial forefoot plantar flexion at initial contact and reducing the propulsive ground reaction forces [37]. The pressures beneath the forefoot and rearfoot in PF patients were notably increased, accompanied by valgus of the rearfoot [38]. Static rearfoot valgus alignment has been widely accepted as the effective predictor of the higher loading rate and plantar loads (higher maximum force and force-time integral) over the rearfoot in acute and chronic PF individuals [24].

With respect to the entire plantar pressure distribution in the midstance phase of walking, the PF athletes’ relative loads were exerted largely on the lateral part of the forefoot; yet, they decreased greatly not merely in the medial part of the midfoot but in the entire rearfoot of both feet as well. The preliminary results of the present study appeared to reconfirmed findings by the related researches who indicated that PF individuals may trigger their feet rolling motion due to foot pain stimuli, and that this syndrome could result not simply in a spontaneous reduction of the plantar load on the rearfoot, but also in greater plantar loads over the forefoot, midfoot and toes [21,22,23] owing to human beings’ protective mechanisms of pains. Based on the findings by Wearing et al. [22], patients with PF make gait adjustments for reducing forces beneath the rearfoot and forefoot of the symptomatic foot, and this increases the toe loading and improves the compensate protective function. Supporting Wearing et al.’s assertions, Sullivan et al. [39] found that people with heel pain reduced maximum forces, peak pressures and force-time integrals over the heel while walking. The results from the present study can be explained by the fact that the rearfoot is the region connected with the pathophysiology and etiology of PF [4]. The PF individuals may experience pains in the calcaneal region which may be attributable to the inflammation or tissue fluid accumulation in the region, and this could result in the increased thickness of the plantar fascia and reduce the capacity of the plantar fascia to support plantar loads over the rearfoot [11,40]. The consequence of this compensation may initiate individuals to spontaneously reduce the plantar load over the rearfoot region of the PF symptomatic foot, and this, in turn, could increase the plantar load over the forefoot [24]. This situation was also observed in the studies of gait tasks in PF conditions [4,23].

As for the entire plantar pressure distribution from the static standing to the midstance phase of walking, the findings of significant difference were observed that the relative loads in the PF group mainly transferred to the lateral part of the entire foot. Notably, however, the male PF athletes’ plantar loads were exerted largely on the lateral forefoot and lateral longitudinal arch of the foot, while the female PF athletes’ plantar loads were mainly distributed on the posterolateral part of the foot. More importantly, the present study is the first attempt to examine and ascertain the notable changes in plantar pressure conversion from the static standing to the midstance phase of walking in hundreds of the PF athletes. A possible explanation for the findings from this study is that individuals with PF usually have static rearfoot valgus alignment and exhibit greater and repetitive plantar loads over the rearfoot [24]. Therefore, there is an increase in the rearfoot pronation and medial forefoot plantar flexion in bipedal static stance for compensatory pain control [24]. However, according to the results from this study, the PF individuals may adjust their gait to control pain, and this could result in the increased plantar loads over the lateral part of the forefoot and reduce the loads over the midfoot and rearfoot regions during the midstance phase of walking. Findings from this study also confirmed that the relative loads of the entire foot were transferred from the medial part to the lateral part during the gait conversion period owing to the individuals’ spontaneous compensatory action to control the painful areas of the symptomatic foot. Apart from this, the plantar loads of the PF males were found to be mainly concentrated on the anterolateral part of the foot. Nevertheless, the plantar loads of the PF females were found to be exerted largely on the posterolateral part of the foot. Due to the differences between the PF males and PF females in terms of their plantar loads, pain assessment of unilateral or bilateral symptomatic foot in different genders of PF participants was worth further exploration. On this basis, the PF athletes’ pain assessment and self-reported health status were conducted to assess their common areas of the musculoskeletal pain and the correlation between the lower extremity musculoskeletal pain and PF.

In the present study, the PF athletes’ common bone pains were mainly located in the calcaneus, the 1st and 2nd plantar metatarsal bone, the 1st and 2nd metatarsophalangeal joint, the lateral ankle joint, the medial knee joint, the femur, the patella, the lateral knee joint, the tibia and the hip joint. Their soft tissue pains occurred frequently in the plantar fascia (medial band, lateral band, and central band), the Achilles tendon, the gastrocnemius, the hamstring muscles, the quadriceps femoris, the plantar plate (abductor halluces and the abductor digiti minimi) and the lower back. The results seemed to resonate with previous studies in which the PF patients noted that their medial plantar heel pains occurred when the Achilles tendon and plantar fascia stretched at the same time during walking, particularly when taking their first steps out of bed in the morning [41]. The results from this study also corresponded to the study by Singh et al. [42]. In their work, Singh et al. observed that the plantar fascia and the Achilles tendon were anatomically connected, and that this was not simply essential for absorbing mechanical shock, stabilizing and preventing the collapse of the longitudinal arch during propulsion, but took the key role in the load distribution of the foot as well. The two structures are closely coherent and show a statistically significant correlation involving the plantar fascia pathology [43]. Research evidence from the related studies verified that the morphological changes of the structures appeared to predispose an individual to the onset of the heel pain and plantar fasciopathy [44]. Apart from this, tightness and excessive stretching of the Achilles tendon have been widely acknowledged as the key and risk factors of plantar fasciopathy [45]. Many studies have been conducted for investigating the biomechanical link between the plantar fascia and the Achilles tendon by measuring the load-deformation, and they also confirmed the positive correlation in podiatric diseases [46,47,48]. In their studies involving the PF patients with Achilles tendon tightness, Tahririan et al. observed that calf stretching was found to be one of the beneficial exercise prescriptions for relieving the PF patients’ pains [49].

Achilles tendon is composed of the gastrocnemius tendon and the soleus muscle. It has been widely accepted that there is a strong link between the gastrocnemius and the plantar fascia [50]. Previous studies showed that the overstretching of Achilles tendon may be attributable to a strong and long-lasting contraction of the gastrocnemius and the soleus muscles, and that this was a reasonable mechanical factor for the excessive weight-bearing of the plantar fascia [50]. Intense strain of the plantar fascia is usually accompanied by an increase in calf muscle tension [43]. Thus, excessive stretching force produced by the gastrocnemius may alter the normal kinematics of ankle motion, and this, in turn, could lead to PF [50]. Findings from the present study revealed that approximately 40% of the PF athletes were assessed to have gastrocnemius pains during the relaxed bipedal static stance. The findings seemed to correspond to the related studies which stated that patients with PF may exhibit greater stiffness on the medial gastrocnemius when their ankles are in a relaxed position, rather than in a neutral position [50]. Supporting the studies by Tahririan et al. [49], many studies went further, asserting that relieving the gastrocnemius stiffness can also simultaneously alleviate the symptoms of PF. This can be achieved via calf stretching exercises, massage and fascia relaxation [41,51,52,53]. Stretching of the gastrocnemius-Achilles tendon complex is widely considered to be an effective exercise prescription for PF and a useful strategy for reducing pains in the foot and ankle, in that stretching may solve PF caused by the tension of the gastrocnemius and intrinsic foot muscles [40,51,52].

Furthermore, the findings from this study revealed that thigh muscle pain was the common pain area in the PF athletes. Previous studies also noted that the muscle strength and reaction time of the thigh, such as the hamstrings and quadriceps muscles which play a vital role in the alteration of lower extremity biomechanics and may contribute to the increased plantar fascia load [54,55,56]. The tightness of hamstring muscles may cause prolonged forefoot loading, which usually results in the increased repetitive stress on the plantar fascia [57,58]. Reaffirming the statements of these studies, findings from the present research stressed that the plantar loads of the PF athletes were particularly inclined to be exerted on the forefoot either in bipedal static stance or during the midstance phase of walking. On this basis, most therapists treated PF patients by means of restoring the flexibility of the posterior thigh muscles such as the gastrocnemius and the hamstrings [38]. Speaking of the impact of PF on the entire lower extremity, Lee et al. [38] went further, arguing that a series of physiological compensatory response of lower limbs, including the quadriceps weakness, delayed reaction time of the hamstring and quadriceps muscles and rapid reaction time of the gastrocnemius muscle, could be found in the affected ankles of PF patients. In addition, compared with healthy individuals, PF patients’ affected ankles tended to exhibit greater plantar loads on the forefoot and rearfoot [38]. Lee et al. also observed that findings from the PF patients displayed a positive correlation with higher incidence of rearfoot valgus [38].

In terms of the relationships between the distribution of plantar load in PF athletes and the pain areas of the musculoskeletal system, previous studies tended to confirm limited ankle dorsiflexion range of motion to be the main cause of PF [59,60]. Based on research evidence, ankle dorsiflexion motion angle less than 10° may cause abnormal compensatory pronation of the subtalar joint during gait, thereby increasing the stress on the plantar fascia [61]. The association between limited ankle dorsiflexion and PF is highly likely to be attributable to soft tissue limitations [59]. Bolgla and Malone [62] reported that PF patients’ foot pronation was commonly attributed to their foot pains which resulted from a stretched plantar fascia. Hence, the patients may use compensatory movements, such as rapid hip flexion, to adjust posture for the relief of their foot pains. Then, the gastrocnemius, hamstrings and quadriceps are highly active in stabilizing the hip and knee joints against gravity and weight [55,63]. Lloyd and Buchanan stated that the co-contraction of the quadriceps and hamstrings directly supported the valgus and varus moments at the knee joint [64]. The valgus and varus moments of the knee joint could lead to foot pronation and supination, respectively, [65,66,67] which may increase plantar fascia stress owing to the increased pressure in the forefoot and the rearfoot [68]. Farhan et al. [69] went further and argued that PF patients often use excessive hip flexion movements to relieve foot pains caused by stretched plantar fascia. Their arguments also correspond to previous studies which observed that hip muscle strength was strongly related to foot pains [70,71]. Based on the arguments by Farhan et al., the quadriceps muscle is a hip flexor, and the weakness of the quadriceps may result from PF patients’ overuse for the relief of foot pains [69,72]. For these reasons, a decrease in the strength of the quadriceps could result in an increase in plantar load and a decrease in the ability to control foot pronation, which usually leads to aggravation of PF symptoms [62,73].

The present study may be limited by the focus on the plantar pressure characteristics of 100 male and 102 female college athletes with PF and aged 20 to 22 years who were college and university students in Taiwan. It is inevitable that the results from the small subset of participants may limit the possibilities for generalization. Even though the process of analysis was conducted by the researcher who specializes in big data analysis, podiatric medical science and plantar pressure measurement, this study might still have deficiencies in analyzing the plantar footprint to a certain degree due to the unavoidable involvement of the researcher’s subjective experiences and non-clinical perspectives within the research context. In addition, this study did not deeply distinguish the severity and grade of the symptomatic feet of the athletes with PF. Therefore, the factors involving the participants’ different levels of PF were not included in the research process within this study. In future studies, it would be worthwhile to further examine the static and dynamic plantar load distributions and the rearfoot postural alignment by considering the severity and grade of symptomatic feet in PF participants.

## 5. Conclusions

In conclusion, the characteristics of the plantar pressure distributions of the athletes suffering from plantar fasciopathy revealed that the higher plantar load mainly distributed on the medial part of the foot, and that this situation was accompanied by a rearfoot valgus in bipedal static stance. The higher plantar loads were mainly exerted on the lateral forefoot and reduced over the rearfoot of the foot during the midstance phase of walking. The transition pattern from the static to dynamic state reflected that the plantar loads of the athletes with plantar fasciopathy were mainly concentrated on the outer part of the entire foot. Notably, however, the plantar loads of the athletes with plantar fasciopathy were exerted largely on the anterolateral part of the foot in males, whilst mainly distributed on the posterolateral part of the foot in females. The lower limb pain profiles seemed to resonate with the plantar pressure distributions, and this could serve as the traceable beginning for the possible links among the pronated low-arched feet, plantar fasciopathy, metatarsalgia, calcanitis and Achilles tendinitis.

## Figures and Tables

**Figure 1 ijerph-18-12942-f001:**
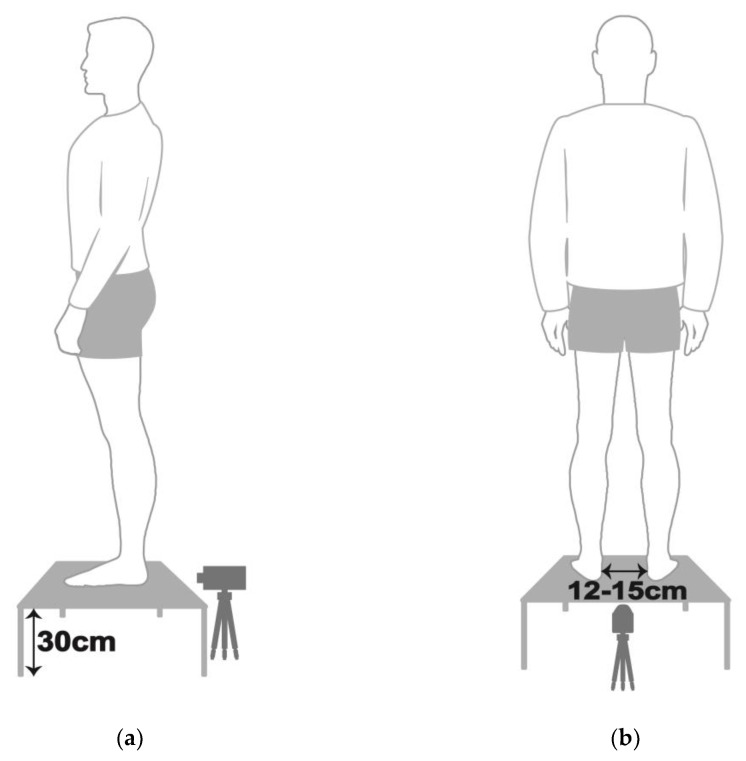
Schematic diagram of each participant with static standing posture in (**a**) lateral view and (**b**) posterior view.

**Figure 2 ijerph-18-12942-f002:**
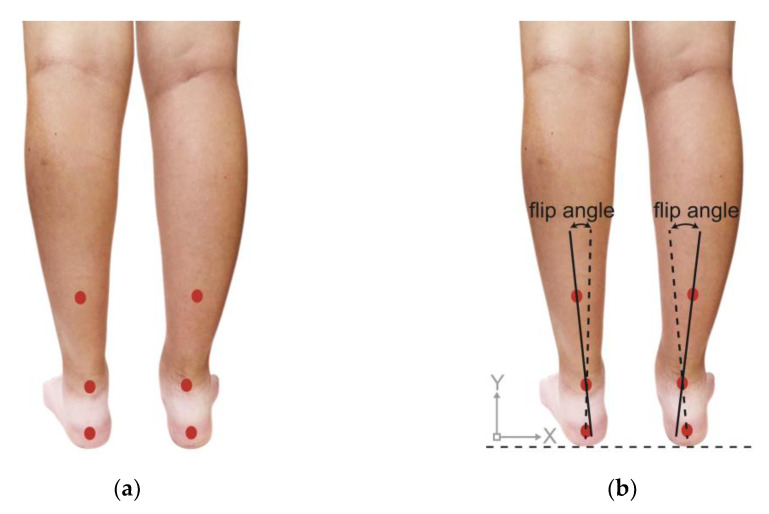
Schematic diagram of (**a**) positioning markers on each participant’s lower back of the leg and (**b**) measurement methods of rearfoot alignment.

**Figure 3 ijerph-18-12942-f003:**
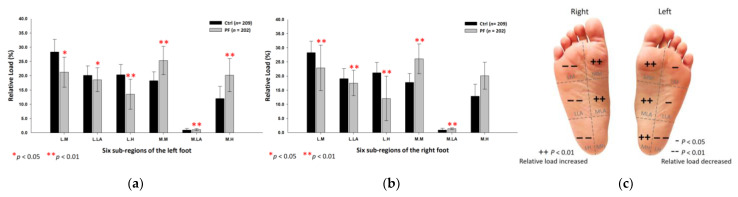
The relative loads of six subregions of (**a**) left and (**b**) right feet in total participants in bipedal static stance. Description of changes in pressure distribution is illustrated in (**c**) plantar diagram. Through the independent sample *t*-test, * *p* < 0.05 and ** *p* < 0.01 are significantly different between the control group and the PF group. The six subregions and their abbreviations are as follows: LH, lateral heel; LLA, lateral longitudinal arch; LM lateral metatarsal bone; MH, medial heel; MLA, medial longitudinal arch; and MM, medial metatarsal bone.

**Figure 4 ijerph-18-12942-f004:**
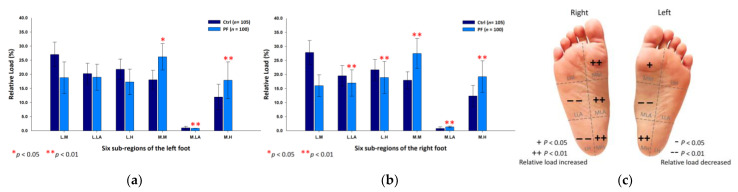
The relative loads of six subregions of (**a**) left and (**b**) right feet in male participants in bipedal static stance. Description of changes in pressure distribution is illustrated in (**c**) plantar diagram. Through the independent sample *t*-test, * *p* < 0.05 and ** *p* < 0.01 are significantly different between the male control group and the male PF group.

**Figure 5 ijerph-18-12942-f005:**
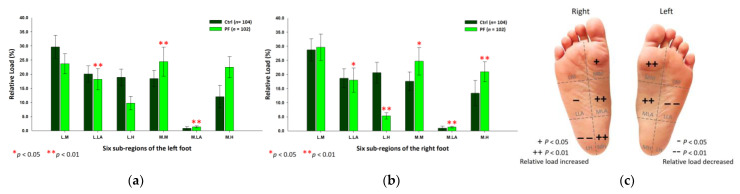
The relative loads of six subregions of (**a**) left and (**b**) right feet in female participants in bipedal static stance. Description of changes in pressure distribution is illustrated in (**c**) plantar diagram. Through in-dependent sample *t*-test, * *p* < 0.05 and ** *p* < 0.01 are significantly different between the female control group and the female PF group.

**Figure 6 ijerph-18-12942-f006:**
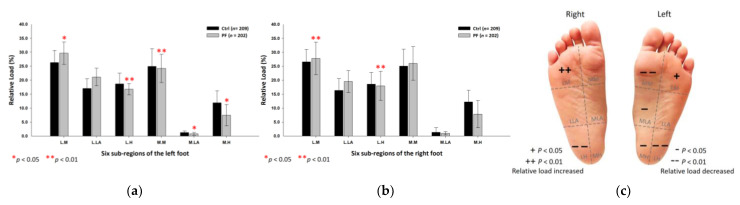
The relative loads of six subregions of (**a**) left and (**b**) right feet in total participants during the midstance phase of walking. Description of changes in pressure distribution is illustrated in (**c**) plantar diagram. Through independent sample *t*-test, * *p* < 0.05 and ** *p* < 0.01 are significantly different between the control group and the PF group.

**Figure 7 ijerph-18-12942-f007:**
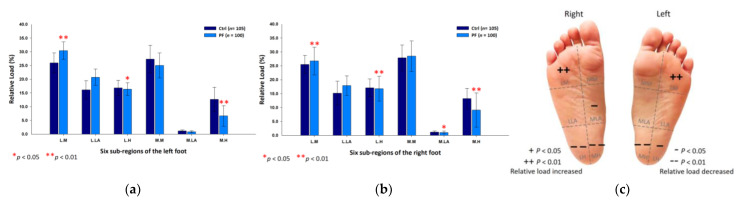
The relative loads of six subregions of (**a**) left and (**b**) right feet in male participants during the midstance phase of walking. Description of changes in pressure distribution is illustrated in (**c**) plantar diagram. Through independent sample *t*-test, * *p* < 0.05 and ** *p* < 0.01 are significantly different between the male control group and the male PF group.

**Figure 8 ijerph-18-12942-f008:**
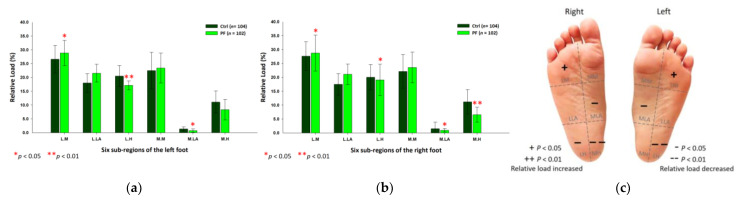
The relative loads of six subregions of (**a**) left and (**b**) right feet in female participants during the midstance phase of walking. Description of changes in pressure distribution is illustrated in (**c**) plantar diagram. Through independent sample *t*-test, * *p* < 0.05 and ** *p* < 0.01 are significantly different between the female control group and the female PF group.

**Table 1 ijerph-18-12942-t001:** Demographic Characteristics of the 411 Study Participants.

	Control Group ^1^			PF Group ^2^		
	Total	Male	Female	Total	Male	Female
Number	209	105	104	202	100	102
Age (years)	20.1 ± 0.9	19.9 ± 0.9	20.2 ± 0.8	21.2 ± 0.8	22.1 ± 0.9	20.2 ± 0.7
Height (cm)	165.7 ± 5.9	168.4 ± 5.2	162.9 ± 5.2	168.6 ± 7.5	170.5 ± 5.8	166.6 ± 4.9
Mass (kg)	61.2 ± 4.7	63.7 ± 4.5	58.6 ± 3.5	64.5 ± 4.8	65.4 ± 4.1	63.5 ± 3.1
BMI	22.3 ± 1.2	22.4 ± 1.0	22.1 ± 1.3	22.7 ± 1.5	22.5 ± 1.6	22.8 ± 1.7

Abbreviation: BMI, body mass index (calculated as the weight in kilograms divided by the square of the height in meters). Note: Data are given as mean ± SD. ^1^ Healthy college students (the control group) were age-matched college and university students. ^2^ College athletes with diagnoses of plantar fasciopathy (the PF group) were college and university students who were qualified first or second-division players from different areas of sports expertise.

**Table 2 ijerph-18-12942-t002:** Arch Index of the Foot in Static Standing.

Gender	Left Foot			Right Foot		
	Control Group	PF Group	*p* Value ^1^	Control Group	PF Group	*p* Value ^1^
Total	0.22 ± 0.08	0.24 ± 0.06	0.593	0.23 ± 0.08	0.25 ± 0.06	0.565
Male	0.23 ± 0.08	0.26 ± 0.07	0.042	0.24 ± 0.08	0.27 ± 0.07	0.037
Female	0.21 ± 0.08	0.22 ± 0.09	0.587	0.22 ± 0.08	0.23 ± 0.08	0.687

Note: Data are given as mean ± SD. ^1^ *p* values were determined by the independent sample *t*-test between the control group (n = 209; male = 105, female = 104) and the PF group (n = 202; male = 100, female = 102).

**Table 3 ijerph-18-12942-t003:** Relative Load of the PF Athletes in Static Standing and Midstance Phase of Walking.

Gender and Region	Static Standing		Midstance Phase of Walking	
	Left Foot	Right Foot	Left Foot	Right Foot
Total (n = 202)				
Lateral Metatarsal bone (LM)	21.27 ± 5.29	22.90 ± 8.06	29.62 ± 4.01 ^2^	27.76 ± 5.84 ^2^
Lateral Longitudinal Arch (LLA)	18.58 ± 4.19	17.50 ± 4.51	21.13 ± 3.12 ^2^	19.50 ± 3.93 ^2^
Lateral Heel (LH)	13.51 ± 5.25	12.06 ± 7.92	16.77 ± 2.01 ^2^	17.95 ± 5.21 ^2^
Medial Metatarsal bone (MM)	25.33 ± 4.97	26.08 ± 5.30	24.19 ± 5.04 ^1^	25.99 ± 6.01
Medial Longitudinal Arch (MLA)	1.12 ± 0.36	1.36 ± 0.30	0.81 ± 0.51 ^2^	0.98 ± 0.57 ^2^
Medial Heel (MH)	20.20 ± 5.78	20.11 ± 4.76	7.48 ± 3.82 ^2^	7.82 ± 4.88 ^2^
Male (n = 100)				
Lateral Metatarsal bone (LM)	18.79 ± 5.61	16.03 ± 3.87	30.42 ± 3.18 ^2^	26.74 ± 4.96 ^2^
Lateral Longitudinal Arch (LLA)	18.95 ± 4.58	16.98 ± 4.67	20.71 ± 2.97 ^2^	17.88 ± 3.51
Lateral Heel (LH)	17.32 ± 4.56	18.91 ± 5.67	16.39 ± 2.31	16.79 ± 4.48 ^2^
Medial Metatarsal bone (MM)	26.21 ± 4.67	27.49 ± 5.32	25.00 ± 4.55	28.47 ± 5.47
Medial Longitudinal Arch (MLA)	0.87 ± 0.02	1.37 ± 0.31	0.85 ± 0.38	1.02 ± 0.49 ^2^
Medial Heel (MH)	17.88 ± 6.52	19.24 ± 5.63	6.63 ± 3.75 ^2^	9.09 ± 6.14 ^2^
Female (n = 102)				
Lateral Metatarsal bone (LM)	23.71 ± 3.57	29.64 ± 4.68	28.85 ± 4.57 ^2^	28.75 ± 6.47
Lateral Longitudinal Arch (LLA)	18.21 ± 3.74	18.00 ± 4.32	21.54 ± 3.22 ^2^	21.09 ± 3.67 ^2^
Lateral Heel (LH)	9.78 ± 2.44	5.35 ± 1.12	17.13 ± 1.59 ^2^	19.08 ± 5.63 ^2^
Medial Metatarsal bone (MM)	24.46 ± 5.13	24.69 ± 4.91	23.40 ± 5.38	23.57 ± 5.53
Medial Longitudinal Arch (MLA)	1.37 ± 0.37	1.35 ± 0.29	0.77 ± 0.61 ^2^	0.94 ± 0.65 ^2^
Medial Heel (MH)	22.47 ± 3.78	20.97 ± 3.54	8.31 ± 3.73 ^2^	6.57 ± 2.70 ^2^

Note: Data are given as mean ± SD. ^1^ *p* < 0.05 versus the same side foot in static standing. ^2^ *p* < 0.01 versus the same side foot in static standing.

**Table 4 ijerph-18-12942-t004:** Static Rearfoot Postural Alignment in PF Athletes.

Gender	Left Foot			Right Foot		
	Control Group	PF Group	*p* Value ^1^	Control Group	PF Group	*p* Value ^1^
Total	4.01 ± 2.52	5.87 ± 3.03	0.037	5.04 ± 3.83	11.85 ± 5.71	0.000
Male	3.64 ± 2.51	5.51 ± 3.76	0.000	4.72 ± 3.38	11.66 ± 6.87	0.000
Female	4.40 ± 2.47	6.23 ± 2.03	0.216	5.39 ± 4.25	12.02 ± 4.31	0.225

Note: Data are given as mean ± SD. ^1^ *p* values were determined by the independent sample *t*-test between the control group (n = 209; male = 105, female = 104) and the PF group (n = 202; male = 100, female = 102).

**Table 5 ijerph-18-12942-t005:** Pain Assessment and Self-Reported Health Status in the PF Athletes.

Bone Pain		Soft-Tissue Pain	
Pain Area	College Athletes with PF (No. [%])	Pain Area	College Athletes with PF (No. [%])
Foot (Calcaneus)	167 (82.7)	Plantar fascia (medial band)	156 (77.2)
Foot (Plantar metatarsal bone 1st & 2nd)	115 (56.9)	Plantar fascia (medial band)	121 (59.9)
Foot (Metatarsophalangeal joint 1st & 2nd)	92 (45.5)	Plantar fascia (medial band)	103 (51.0)
Lateral ankle joint	73 (36.1)	Achilles tendon	88 (43.6)
Medial knee joint	61 (30.2)	Gastrocnemius	81 (40.1)
Femur	56 (27.7)	Hamstring muscles	76 (37.6)
Patella	55 (27.2)	Quadriceps femoris	69 (34.2)
Lateral knee joint	47 (23.3)	Plantar plate (Abductor hallucis)	41 (20.3)
Tibia	38 (18.8)	Plantar plate (Abductor digiti minimi)	35 (17.3)
Hip joint	27 (13.4)	Lower back	30 (14.9)

## Data Availability

The datasets generated and/or analyzed for the present study are available from the corresponding author upon reasonable request.

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
