# Peer review of "Relationships between Plantar Pressure Distribution and Rearfoot Alignment in the Taiwanese College Athletes with Plantar Fasciopathy during Static Standing and Walking"

_ijerph, 2021, doi:10.3390/ijerph182412942_

Round 1

Reviewer 1 Report

The pressure platform used should have reliability and validity to extrapolate the data obtained.

Author Response

Dear Reviewer,

Coauthors and I appreciated the reviewers’ constructive suggestions and constructive comments on our manuscript (ID: ijerph-1441844). The suggestions and comments are helpful for improving our manuscript. We are submitting the revised version of the manuscript with our responses to the suggestions and comments by the reviewers.

Our responses to each suggestion and comment are as follows, and they are also presented in red text with a grey background color in the revised manuscript:

Reviewer 2 Report

Comment to author

This is a valuable study of plantar loading in a large population.

The research topic is interesting, but I have some questions about this study, which I would like to get answers from the author.

Background

Line 79

A few lines of text from Line 79 should be stated in the discussion.

Line 111

Why did you define BMI as 18.5~24.0?

The load on foot alignment is likely to be different between a BMI of 18.5 and 24.0, and the reason for defining BMI in such a range should be stated.

Line 118

Exclusion criteria should be listed within the text of the Method, rather than in the caption.

Methods

"both feet" in Line 256. Does this mean that both lower limbs were included in the study? If you are measuring for both lower limbs, the data includes the healthy lower limb of the PF patient, which may be an inappropriate target for data measurement.

Line 182

Regarding the method of measuring rearfoot static angle, I think it is the method described in Fig. 7 in the Result session, but an explanatory diagram should be inserted in the Method as well.

Line 188

Since there is no diagram explaining a solid line or a dot line near this text, it seems difficult for the reader to imagine it.

Line 193

Is the classification of Normal foot, varus, and valgus used as a result of this study? The reader may be confused because it gives the impression that this classification is used in the analysis.

Line 211

What data items are included in the self-reported health status? Please provide details.

Result

Line 293 299 303

The differences in the attributes of the results shown in Figs. 1-3 should be mentioned in the captions. (e.g. fig 1: total subject, fig 2: male subject, fig 3: female subject)

Discussions

Line 393

If I may judge from your research data, the normal value for AI is 0.22-0.23, not 0.21-0.26.

Line 418

Regression analysis was not conduct in this study, and it is preferable to avoid such expressions.

Author Response

(The authors gave the same response as above.)

Reviewer 3 Report

The work addresses a topic of interest to consider for publication, but has great methodological shortcomings to be accepted for publication.

  • The work does not have a designed goal. From line 76 of the introduction, the authors make a small summary about the work, this should be eliminated and replaced by the objective or objectives of the 
  • The methodology is not ordered clearly, concisely and chronologically.
  • Recruitment, group formation and inclusion and exclusion criteria are unclear.
  • The study was not approved by an ethics committee.
  • They do not justify the procedure of diagnosis of pathology.
  • The analysis of the plantar footprint presents great deficiencies, possibly because it was carried out by non-specialist personnel in the area such as podiatrists.
  • The results do not show in a clear and direct way.
  • The conclusions do not clearly answer the objective, it must be because it is not formulated.

Author Response

(The authors gave the same response as above.)

Round 2

Reviewer 1 Report

Currently the term plantar fasciitis is displaced and it is preferred to use the term plantar fasciopathy that better describes the pathology that develops in the injury

Author Response

Dear Reviewer,

Coauthors and I appreciated the reviewers’ constructive suggestions and constructive comments on our manuscript (ID: ijerph-1441844). The suggestions and comments are helpful for improving our manuscript. We are submitting the revised version of the manuscript with our responses to the suggestions and comments by the reviewers.

Our responses to each suggestion and comment are as follows, and they are also presented in blue text with a grey background color in the revised manuscript:

Reviewer 2 Report

Thank you for responding to my comment.

Reviewer 3 Report

Congratulations to the authors for the work done to improve the work, but I still observe small nuances that they must take into account.

  • In the introduction they keep making a job description, I don't think it's necessary.... A good description of the target is enough.
  • Excellent changes in the methodology are observed, achieving the objective of the same... be able to replicate the study. But I think that some sections should be reviewed to expose it in a more  synthesizedway... . the authors extend too much.
  • Excellent presentation of the results.
  • The conclusions respond to the objective of the work based on the results obtained, but I recommend that in this section no abbreviations are used.

Author Response

(The authors gave the same response as above.)
